# Exploring the Therapeutic Potential of *Ampelopsis grossedentata* Leaf Extract as an Anti-Inflammatory and Antioxidant Agent in Human Immune Cells

**DOI:** 10.3390/ijms25010416

**Published:** 2023-12-28

**Authors:** Arthur Chervet, Rawan Nehme, Caroline Decombat, Lucie Longechamp, Ola Habanjar, Amandine Rousset, Didier Fraisse, Christelle Blavignac, Edith Filaire, Jean-Yves Berthon, Laetitia Delort, Florence Caldefie-Chezet

**Affiliations:** 1Université Clermont-Auvergne, INRAE, UNH, Unité de Nutrition Humaine, CRNH-Auvergne, 63000 Clermont-Ferrand, France; arthur.chervet@doctorant.uca.fr (A.C.); rawan.nehme@doctorant.uca.fr (R.N.); caroline.decombat@uca.fr (C.D.); lucie.longechamp@uca.fr (L.L.); ola.habanjar@uca.fr (O.H.); didier.fraisse@uca.fr (D.F.); edith.filaire@uca.fr (E.F.); florence.caldefie-chezet@uca.fr (F.C.-C.); 2Greentech, Biopôle Clermont-Limagne, 63360 Saint-Beauzire, Francejeanyvesberthon@greentech.fr (J.-Y.B.); 3Centre Imagerie Cellulaire Santé, Université Clermont Auvergne, 63000 Clermont-Ferrand, France; christelle.blavignac@uca.fr

**Keywords:** *Ampelopsis grossedentata*, antioxidant, inflammation, macrophage polarization

## Abstract

Inflammation is a vital protective response to threats, but it can turn harmful if chronic and uncontrolled. Key elements involve pro-inflammatory cells and signaling pathways, including the secretion of pro-inflammatory cytokines, NF-κB, reactive oxygen species (ROS) production, and the activation of the NLRP3 inflammasome. *Ampelopsis grossedentata*, or vine tea, contains dihydromyricetin (DHM) and myricetin, which are known for their various health benefits, including anti-inflammatory properties. Therefore, the aim of this study is to assess the impact of an extract of *A*. *grossedentata* leaves (50 µg/mL) on inflammation factors such as inflammasome, pro-inflammatory pathways, and macrophage polarization, as well as its antioxidant properties, with a view to combating the development of low-grade inflammation. *Ampelopsis grossedentata* extract (APG) significantly decreased ROS production and the secretion of pro-inflammatory cytokines (IFNγ, IL-12, IL-2, and IL-17a) in human leukocytes. In addition, APG reduced LPS/IFNγ -induced M1-like macrophage polarization, resulting in a significant decrease in the expression of the pro-inflammatory cytokines TNF-α and IL-6, along with a decrease in the percentage of M1 macrophages and an increase in M0 macrophages. Simultaneously, a significant decrease in NF-κB p65 phosphorylation and in the expression of inflammasome genes (*NLRP3, IL-1β* and *Caspase 1*) was observed. The results suggest that *Ampelopsis grossedentata* could be a promising option for managing inflammation-related chronic diseases. Further research is needed to optimize dosage and administration methods.

## 1. Introduction

Inflammation is a natural response of the body to physical, pathogenic, and chemical attacks [1]. When the reaction is appropriate, controlled, and time-limited, it is protective and essential to our survival. However, persistent chronic inflammation, resulting from an imbalance between pro- and anti-inflammatory factors, has been shown to be harmful and leads to the development of many diseases [2]. An essential component of the inflammatory response is the activation of pro-inflammatory cells—in particular, phagocytic mononuclear cells and neutrophils, during the acute inflammatory phase. Indeed, in response to primary inflammatory stimuli such as lipopolysaccharide (LPS), these cells will be activated and secrete a wide variety of cytokines and pro-inflammatory mediators, including tumor necrosis factor (TNF-α), interleukin (IL)-6, IL-1β, prostaglandin E2 (PGE2) and reactive oxygen species (ROS) [3,4,5]. These immune cell secretions mediate inflammation by interacting with Toll-like receptors (TLRs), specific receptors for TNF-α (TNFRs), IL-1 (IL-1R), and IL-6 (IL-6R). Receptor activation triggers important intracellular signaling pathways, such as nuclear factor kappa-B (NF-κB), which is strongly influenced by ROS production [6]. Indeed, several studies have shown that ROS can induce the phosphorylation of IκB, leading to its degradation, which allows NF-κB to move into the cell nucleus and activate the expression of inflammation-specific genes. ROS can also directly activate protein kinases involved in the NF-κB signaling pathway, such as IKK kinase, which leads to the activation of NF-κB [7]. Furthermore, an important player in the regulation of inflammation is NLRP3 (NOD-like receptor family, pyrin domain containing 3). NLRP3 is an intracellular sensor that recognizes various danger signals and forms the NLRP3 inflammasome—a multiprotein complex involved in inflammation [8]. The activation of the NLRP3 inflammasome leads to the activation of Caspase-1, which processes and releases pro-inflammatory cytokines, including IL-1β and IL-18. The NLRP3 inflammasome can be activated by a range of stimuli, including pathogen-associated molecular patterns (PAMPs) and danger-associated molecular patterns (DAMPs), such as bacterial components (LPS), endogenous danger signals (ATP), uric acid crystals, and ROS [8].

Since the 2000s, many studies have focused on the use of plants for their beneficial effects on human health, particularly through various compounds such as flavonoids. It has been shown that flavonoids exert an anti-inflammatory effect by acting on several mechanisms [9]. Firstly, they can reduce the production of pro-inflammatory molecules, such as cytokines and prostaglandins, by inhibiting the enzymes responsible for their production [10]. For example, quercetin can inhibit the enzyme cyclooxygenase-2 (COX-2) involved in the production of prostaglandins [11]. In addition, flavonoids can modulate the activation of pro-inflammatory immune cells, such as macrophages. They can inhibit the activation of signaling pathways that lead to the production of pro-inflammatory cytokines [12]. Flavonoids also exert an antioxidant effect by neutralizing free radicals, which may help reduce inflammation. Free radicals, produced during inflammatory processes, can damage surrounding cells and tissues, thus contributing to the perpetuation of the inflammatory process [13]. Finally, flavonoids can inhibit the expression of adhesion molecules, which are involved in the adhesion of immune cells to surrounding tissues [14]. This may reduce the infiltration of immune cells into tissues, which may help to reduce inflammation.

*Ampelopsis grossedentata* (Hand.-Mazz.) is a traditional herb that s widely used in traditional Chinese medicine. This plant, also known as vine tea, is generally used as an herbal tea for the treatment of coughs, colds, fever, sore throat, vomiting, and chronic nephritis thanks to its rich flavonoid content. Phytochemical studies have shown that dihydromyricetin (3,5,7,3′,4′,5′-hexahydroxy-2,3-dihydroflavonol, DHM) and myricetin are the two main flavonoids in *A*. *grossedentata* [15]. Many scientific studies have shown that DHM has a variety of biological functions, including anti-inflammatory, antioxidant, anticancer, and antidiabetic activities [16,17,18,19]. However, these anti-inflammatory properties have never been examined on *A*. *grossedentata* leaf extract, and no advanced studies on human immune cells have been conducted.

In the present study, we examined the effects of an *A*. *grossedentata* leaf extract (APG) on several inflammation actors such as inflammasome, pro-inflammatory signaling pathways, and macrophage polarization, and also its antioxidant properties (production of ROS) on human immune cells. The evidence of the efficacy of APG in reducing inflammation could lead to its having a real beneficial impact in curbing the progression of several diseases associated with these processes. 

## 2. Results

### 2.1. Quantification of Dihydromyricetin (DHM) Content with HPLC-UV

DHM is the largely predominant peak in the phenolic profile produced by HPLC. The quantification showed a content of 31.2% DHM in the dry extract (Appendix A).

### 2.2. Ampelopsis Grossedentata Extract Inhibited ROS Production via Blood Leukocytes

To investigate the potential antioxidant effect of APG, we examined its effect on blood leukocyte ROS production triggered by PMA. The incubation of leukocytes with APG significantly inhibited the production of ROS in a dose-dependent manner (Figure 1A) (−33 ± 4%, −41 ± 3.3%, −49 ± 3.3%, and −51 ± 4.4% at 10, 25, 50, and 100 µg/mL of APG, respectively). This effect was not the consequence of a decrease in cell viability. On the contrary, viability was increased at high APG concentrations (Figure 1B). In addition, to assess the radical scavenging activity of APG, we used the DPPH assay. As shown in Figure 1C, the concentration for scavenging 50% DPPH radicals (IC50) by APG extract was 3.9 µg/mL.

### 2.3. Ampelopsis Grossedentata Extract Significantly Decreased the Production of Pro-Inflammatory Cytokines in PHA-Stimulated PBMCs

Phytohemagglutinin (PHA) is a lectin that binds non-specifically to the sugars on glycosylated surface proteins, including the T-cell receptor (TCR). To better understand whether cytokine production was influenced by APG, PBMCs, both with and without stimulation with PHA, they were treated with 50 µg/mL of APG, and the secretion of 10 cytokines was measured in the cell supernatants using the Luminex Bio-Plex 200 System. Treatment with APG induced a significant decrease in IFNγ, IL-12, IL-17a, and IL-2 (−99.3 ± 0.6%, −61.5 ± 20.4%, −94 ± 3.6%, and −79.8 ± 14%, respectively) compared to the control (Figure 2A). For IL1-β, MIP-1α and IL-23, a non-significant decrease was observed (−24.3 ± 16.5%, −27.2 ± 20.5% and −38 ± 5.2%, respectively). The treatment did not seem to have any impact on IL-6, IL-8, or TNF-α secretion by PBMCs.

### 2.4. Ampelopsis Grossedentata Extract Inhibited the Activation of NF-κB in LPS-Stimulated PBMCs

To assess whether the inhibitory effect of APG on pro-inflammatory cytokines was mediated through NF-κB, we examined the effect of APG on NF-κB p65 phosphorylation. As demonstrated in Figure 2C, LPS significantly enhanced p65 phosphorylation compared to unstimulated PBMCs. The treatment with APG on LPS-stimulated PBMCs induced a significant decrease in NF-κB p65 phosphorylation (−36.3 ± 10.4%). APG extract on unstimulated cells also induced a decrease in NF-κB p65 phosphorylation compared to LPS-stimulated PBMCs and had no effect compared to unstimulated PBMCs.

### 2.5. Ampelopsis Grossedentata Extract Inhibited the NRLP3 Gene-Expression Pathway and COX-2 in LPS-Stimulated Blood Leukocytes

Based on the results obtained for ROS production, we investigated whether APG had an effect on the expression of the antioxidant enzymes Heme oxygenase-1 (*HO-1*) and Glutathione Peroxidase 1 (*GPX1*), as well as the transcription factor *Nrf2*. The results showed that there was no significant modification in the expression of these genes in any of our conditions compared with LPS-treated cells (Figure 3A). Then, to see whether the NLRP3 pathway components were affected by APG extract, blood leukocytes were stimulated with LPS in the presence or absence of APG for a period of 4 h, and the gene expression of *NLRP3*, *IL-1β*, and *Caspase-1* was measured. As expected, LPS significantly induced the expression of *NLRP3*, *IL-1β*, and *Caspase-1* (Figure 3B), while APG greatly decreased the expression of all the studied genes (RQ = 0.60 ± 0.23; 0.60 ± 0.26 and 0.51 ± 0.20 for *NRLP3*, *IL-1β* and *Caspase-1*, respectively). Furthermore, APG extract had no effect on unstimulated blood leukocytes compared to untreated cells. Finally, we investigated the effect of APG on *COX-2* gene expression and PGE2 secretion by LPS-stimulated blood leukocytes. LPS considerably increased *COX-2* expression (Figure 3C), and APG extract significantly decreased it in LPS-stimulated cells (RQ = 0.54 ± 0.30). As for the results obtained on unstimulated cells treated with APG, no difference was noted compared with untreated cells (0.08 ± 0.04 vs. 0.10 ± 0.07). In a non-inflammatory situation, APG had no effect on *COX-2* expression. The results obtained for *COX-2* were not correlated with those of PGE2. Indeed, APG did not significantly reduce PGE2 secretion on stimulated cells compared to the positive control (1830.9 ± 211.9 pg/mL vs. 2140.8 ± 370.9 pg/mL) (Figure 3D).

### 2.6. Ampelopsis Grossedentata Extract Significantly Decreased the Gene Expression of Inflammatory Cytokines in M1-Type Macrophages

To see whether APG was able to act on macrophages (another key player in innate immunity), THP-1 cells were treated with APG (50 µg/mL) during the activation of THP-1 cells into M0 macrophages or during the polarization into the M1-type. APG had no effect on the expression of the studied genes (*TNF-α*, *IL-6*, *IL-8*, *IL-1β*) in activated macrophages, with the exception of *CXCL10*, though this is not significant (Figure 4). When we evaluated the impact of APG during the polarization of macrophages into M1-type, APG significantly decreased the expression of *TNF-α* (RQ = 0.57 ± 0.13) and *IL-6* (RQ = 0.20 ± 0.26). A non-significant decrease in *IL-8* (RQ = 0.81 ± 0.43), *IL-1β* (RQ = 0.78 ± 0.26), and *CXCL10* (RQ = 0.52 ± 0.21) expression was observed. These results were confirmed by flow cytometry. As shown in Figure 5A, the addition of APG during macrophage activation (without LPS or IFNγ) did not significantly alter the proportion of M0 or M1 macrophages (Figure 5B). On the contrary, APG was able to significantly reduce the proportion of the M1 macrophages (70.29% ± 2.71 vs. 90.72% ± 0.47) and increase the proportion of M0 (9.27 ± 0.45 vs. 29.70 ± 2.71) (Figure 5B).

## 3. Discussion

Chronic low-grade inflammation is now recognized as a risk factor for metabolic illnesses such as obesity, type-2 diabetes, and cardiovascular disease [20]. With plants being the primary source of novel active substances, research into medicinal plants with anti-inflammatory potential is of particular interest. In our study, we focused on an extract of the Chinese medicinal plant *A*. *grossedentata*. In this context, we investigated the effect of *A*. *grossedentata* extract (APG) on several inflammatory pathways in human immune cells, with the aim of preventing this process.

*A*. *grossedentata* is a plant of considerable interest, known for its medicinal properties and its composition of bioactive compounds—notably a flavonoid called dihydromyricetin (DHM). The HPCL phenolic profile of *A*. *grossedentata* shows a very largely predominant peak, the DMH. The identification of this compound was carried out according to Gao et al. [21]. Our extract, with a high DHM content (312 mg/g of dry extract), is in agreement with other studies described in the literature [22]. DHM is being explored for its several potential health advantages, including anticancer, antioxidant, anti-inflammatory, and antibacterial effects [16,17,23,24]. As an antioxidant, it contributes to the neutralization of free radicals and the reduction of oxidative stress. Its anti-inflammatory qualities help in the modulation of inflammatory pathways and the reduction in pro-inflammatory cytokine production. Nevertheless, these properties have never been demonstrated on a leaf extract or in a study on human immune cells. Additionally, DHM has anticancer properties, decreasing cancer-cell proliferation and metastasis [18]. Furthermore, it is being investigated for its hepatoprotective properties and its capacity to protect liver cells from harm and improve liver function [25,26]. However, chemical instability and poor bioavailability are two major drawbacks to the use of DHM. Because DHM is only soluble in hot water and ethanol and poorly soluble in water at room temperature, it has limited membrane permeability and bioavailability [27,28]. These are determinant factors that limit the pharmacological activity and clinical applicability of DHM.

The activation of NF-κB is a tightly regulated process that is crucial in immune response and inflammation. NF-κB is a heterodimer made up of p50 and p65/RelA subunits. In its inactive state, NF-κB is sequestered in the cytoplasm through its association with inhibitory proteins IκBs. Upon stimulation by various factors such as pro-inflammatory cytokines, pathogens, or cellular stress, specific signaling pathways are initiated. This leads to the activation of the inhibitor of κB kinase (IKK) complex, which phosphorylates the IκBs. Phosphorylation marks the IκBs for proteasomal degradation, allowing NF-κB to be released from its inhibitory complex. This permits the liberated NF-κB dimers to translocate into the nucleus, where they bind to the promoter of target genes and activate transcription [29]. In our investigation, we observed that APG lowered NF-κB pathway activation by decreasing the phosphorylation of the p65 subunit by 36% (Figure 6). DHM is already known to downregulate the expression of the NF-κB signal. Indeed, Hou et al. showed that DHM inhibited p-NFκB p65 and p-IκBα, indicating NF-κB activation in RAW 264.7 cells [30]. Wang et al. also showed that DHM inhibited macrophage activation by suppressing p65 phosphorylation, IKKα/β phosphorylation, and IKKβ activity [31]. The NF-κB pathway plays a crucial role in the transcription of genes encoding pro-inflammatory cytokines such as TNF-α, IL-1β, and IL-6. When cells are exposed to inflammatory stimuli, the activation of NF-κB leads to overexpression of these pro-inflammatory cytokines, which play important roles in amplifying the inflammatory response, regulating immune cell chemotaxis, and modulating immune responses. Our study revealed in PHA-stimulated PBMCs that APG inhibited the production of several inflammatory cytokines, IFNγ, IL-12, IL-17a, and IL-2 by 99%, 61%, 94%, and 80%, respectively (Figure 6). A few studies have found that DHM reduces the production of TNF-α, IL-6, and IL-1β. The first study, by Hou et al., showed that DHM reduced the production of these cytokines in both LPS-inflamed RAW 264.7 and LPS-stimulated BALB/c mice serum [30]. The effects of DHM on TNF-α were only seen at the highest concentrations (460 mg/kg on mice and 300 µM on cells). Chu et al. observed that DHM reduced the levels of TNF-α, IL-6, and IL-1β (from 50 mg/kg for TNF-α to 20 mg/kg per day for IL-6 and IL-1β) in complete Freund adjuvant rats to produce rheumatoid arthritis [32]. In another study, which is closer to ours in terms of tested extract, Wang et al. showed decreases in the quantity of IL-6 and IL-1β (50 mg/kg for IL-6 and 100 mg/kg for IL-1β) LPS-stimulated ICR mice serum [33]. The disparity in TNF-α, IL-6, and IL-1β findings could be attributed to the cell stimulant used. LPS is a component of Gram-negative bacteria’s cell walls. Immune cells—mostly macrophages—recognize it through the TLR4 receptor. TLR4 activation by LPS initiates a signaling cascade—most notably involving the NF-κB pathway—which results in the release of pro-inflammatory cytokines such as TNF α, IL-1β, and IL-6 [34]. On the other hand, PHA is a plant lectin that causes an inflammatory response mostly mediated by T cells. It stimulates T cells by binding to their TCRs (T-cell receptors), causing them to multiply and produce cytokines—notably pro-inflammatory cytokines such as IL-2, IFNγ, IL-12, and IL-17 [35,36]. Furthermore, the NF-κB pathway promotes the expression of COX-2 and, as a result, the generation of PGE2 [37]. COX-2 is an enzyme involved in prostaglandin synthesis, and PGE2 is a pro-inflammatory mediator. PGE2 promotes vasodilation, increases vascular permeability, and sensitizes pain receptors, all of which contribute to the inflammatory response [38]. We found that APG reduced *COX-2* expression in LPS-stimulated blood leukocytes but had no effect on PGE2 release (Figure 6). These results are comparable to those found by Qi et al., who observed no effect of DHM at 50 µg/mL on 16 h LPS-stimulated RAW264.7 cells [39]. Unlike our investigations and the study conducted by Hou et al., Qi et al. found no effect of DHM on COX-2 protein levels [30].

In addition to being a key mediator of immunity, NF-κB plays a critical role in priming the NLRP3 inflammasome [40]. When NLRP3 is activated, it recruits Caspase-1, which, in turn, facilitates the conversion of pro-IL-1β into its mature and active form, IL-1β. Thus, Caspase-1 plays a crucial role in the maturation and secretion of IL-1β [41]. Several studies have demonstrated that Chinese medicinal herbs can influence the NLRP3 inflammasome. Pretreatment with *Cinnamomum osmophloeum* essential oil reduced the expression of Caspase-1 and NLRP3 in the intestinal mucosa and also reduced serum levels of IL-1β and IL-18 [42]. Another study found that Licochalcone B (the primary component of licorice) has a particular inhibitory impact on the NLRP3 inflammasome in mice and human immune cells [43]. Our study showed that in LPS-stimulated blood leukocytes, APG reduced the gene expression of NLRP3 inflammasome, as well as *Caspase-1* and *IL-1β* (Figure 6). Shi et al. found that DHM administration reduced IL-1β protein levels in chicken serum, as well as IL-1β, NLRP3, Caspase-1 mRNA, and protein, in chickens with LPS-induced hepatotoxicity [44]. M1 macrophages exhibited pro-inflammatory cytokine production and microbiocidal action [45]. However, persistent overactivation of the pro-inflammatory activities of M1 macrophages led to chronic inflammation and the development of autoimmune disorders [46]. Various medicinal plants have been studied for their ability to inhibit the polarization of M1 macrophages. Curcumin reduced the polarization of M1 macrophages induced by LPS/IFNγ from THP-1 in a dose-dependent manner, according to Zhou et al. [47]. A second study, by Shabani et al., found that resveratrol reduced the polarization of M1 macrophages in the skeletal muscle tissue of mice fed a high-fat diet [48]. Our findings revealed that APG greatly lowered by 20% the polarization of M1 macrophages. When our extract was exposed to M0 without LPS/IFNγ, there was no increase in M1 polarization compared to the negative control. In a non-inflammatory situation, APG extract induced no pro-polarizing effect. The modulation of macrophage (M1/M2) polarization using a natural extract could be a novel therapeutic tool in the fight against chronic and other diseases.

On the other hand, oxidative stress is closely linked to inflammation. ROS production plays a central role in the progression of many inflammatory diseases. They function as both a signaling molecule and an inflammatory mediator. ROS are generated by cells involved in host defense, such as polymorphonuclear neutrophils (PMNs) [49]. Flavonoids are known to have antioxidant properties—notably, the ability to capture ROS, inhibit oxidases which produce superoxide anions, chelate trace metals, and activate antioxidant enzymes [50]. Our study showed that ROS production decreased in a dose-dependent manner in PMA-stimulated blood leukocytes. In parallel, we found that the concentration for scavenging 50% DPPH radicals (IC50) by APG was 3.9 µg/mL. The results concerning the scavenging effect of APG were similar to those found for pure DHM. Indeed, Xie et al. showed an EC50 of 3.24 μg/mL for pure DHM [51]. Two other studies carried out on extracts of *Ampelopsis grossedentata* showed divergent results, probably due to the extraction method. Indeed, the first study conducted by Zhang et al. on vine tea powders extracted with petroleum ether showed an IC50 of 15.35 µg/mL, which is similar to our results [52]. On the other hand, Wang et al. showed an IC50 of around 0.4 mg/mL for a commercial water extract of *Ampelopsis grossendentata* [33]. Regarding ROS production, a study by Qi et al. also showed a dose-dependent reduction in ROS production in LPS-stimulated RAW 264.7 [39]. In this study, we also wanted to see the impact of APG extract on the gene expression of *Nrf2*, *HO-1*, and *GPX1* on LPS-stimulated blood leukocytes, which are involved in antioxidant responses. In response to a variety of stimuli, Nrf2 binds to the ARE and enhances the expression of several antioxidant genes [53]. Using Western blotting, Luo et al., showed an increase in Nrf2/HO-1 on DMH-pre-treated (40 µM, 0–4–8–12 h) and ox-LDL-induced HUVEC injury [54]. Hu et al., also using Western blotting, found an increase in Nrf2/HO-1 at 24 h in palmitic acid-stimulated HUVECs pretreated for 12 h with 0.1, 0.5, and 1 µM DMH [55]. In contrast to these studies, we observed no increase in the expression of Nrf2, HO-1, and GPX1 on LPS-stimulated blood leukocytes after 4 h in the presence of APG and LPS. It might be interesting to look at the impact of the extract after 12 h pretreatment.

Our results clearly suggest, for the first time using human immune cells, that *A*. *grossedentata* is a promising agent for controlling inflammation and preventing inflammation-related chronic diseases. To our knowledge, no previous research has been conducted on the effect of *A*. *grossedentata* on the various components of inflammation in human leukocytes. Nonetheless, the study is limited by the question of bioavailability and adequate dosage. Indeed, there has been some bioavailability and pharmacokinetic research on DHM, but none on *Ampelopsis grossendatata* extract, and DHM investigations demonstrated very poor bioavailability. Indeed, Ruan et al. demonstrated in an in vitro investigation utilizing Caco-2 cells that the molecule’s bioavailability was less than 10% [56]. Furthermore, Liu et al. showed in an in vivo investigation in rats that the maximal concentration (Cmax) of DHM was only 21.63 ± 3.62 ng/mL after 2.67 h of treatment with an oral dosage of 20 mg/kg. After intravenous administration of 2 mg/kg DHM, the Cmax was 165.67 ± 16.35 ng/mL, the half-life was 2.05 ± 0.52 h, and the residence time was only 2.62 ± 0.36 h, according to the same study. In rats, the bioavailability was found to be 4.02% [27]. To replicate the in vitro effects, physicochemical characteristics of this molecule would need to be considerably modified in order to make it more bioavailable and hence enhance its effects in vivo. In conclusion, our findings suggest that *A*. *grossedentata* could be a promising option for managing inflammation and preventing the development of associated diseases such as chronic inflammatory bowel disease, metabolic diseases including type-2 diabetes, and skin diseases, for example. Further studies are required to determine the other main components of our extract and to investigate bioavailability assays.

## 4. Materials and Methods

### 4.1. Quantification of Dihydromyricetin (DHM) Content using HPLC-UV

Dihydromyricetin (DHM) was analyzed using high-performance liquid chromatography (HPLC) linked to a 200–400 nm diode array detector (Agilent, Santa Clara, CA, USA). Analysis was performed with a Lichospher^®^ C18 (125 mm × 4 mm × 5 µm; Merck, Rahway, NJ, USA) column at 20 °C and a flow rate of 1 mL/min. The initial mobile phase was composed of 90% acidified water with 1% orthophosphoric acid and 10% acetonitrile. A linear gradient was applied: 0–15 min, 10–15% B; 15–25 min, 15% B; 25–40 min, 15–20% B; 40–50 min, 20–40% B; 50–60 min, 40–60% B. A 10 µL measure of a 50 mg/mL (water/methanol, 5/95, *v*/*v*) extract solution was injected. DHM has been identified according to its retention time and DAD-UV-Vis (200–400 nm) characteristics in comparison to commercial standards or data reported in the literature. Quantification was performed using UV at 290 nm with a DHM standard (PhytoLab, Vestenbergsgreuth, Germany).

### 4.2. Preparation of Ampelopsis Grossedentata Extract

The leaves of *A*. *grossedentata* (Hunan Province, China) were crushed into 0.1 to 0.5 mm pieces after being sun-dried. Then, 10 g of dried plants were extracted in 200 mL of a 50/50 mixture of ethanol and water while being shaken on a rotary shaker for 4 h at room temperature. After being concentrated in a rotary evaporator (at 40 °C), the extract yielded 3.4 g of dry powder, containing 31.2% DHM. After that, the powder was dissolved in DMSO and kept at −20 °C. By diluting the extract in complete culture medium just before use, the final concentrations were obtained: 0, 10, 25, 50, and 100 µg/mL.

### 4.3. Blood Leukocyte Preparation

Blood was collected from healthy human volunteers (3 volunteers, Etablissement Français du Sang, EFS, Clermont-Ferrand, France). Donors gave their written informed consent for the use of blood samples for research purposes under EFS contract no. 16-21-62 (in accordance with articles L1222-1, L1222-8, L1243-4 and R1243-61 of the French Public Health Code). Whole blood leukocytes were obtained via hemolytic shock using ammonium chloride solution (NH4Cl 115 µM; NaHCO_3_ 12 µM, EDTA 0.01 µM), followed by centrifugation (1300 rpm, 10 min), and they were washed and suspended in Roswell Park Memorial Institute 1640 Medium (RPMI-1640, Gibco, Thermo Fisher Scientific, Waltham, MA, USA) and supplemented with fetal bovine serum (FBS, 10%) (Eurobio Scientific, Saclay, France), gentamicin (50 μg/mL), and glutamine (Gln, 2 mM) (Thermo Fisher Scientific).

### 4.4. PBMC Preparation from Human Blood

Blood buffy coats were collected from healthy human volunteers (3 volunteers, EFS, Clermont-Ferrand, France) and carefully layered on a gradient of Ficoll–Histopaque^®^-1077 (Sigma-Aldrich, St. Louis, MO, USA). The first layer of plasma was aspirated after centrifugation (1500 rpm, 40 min at 25 °C), revealing a phase of monocytes and lymphocytes (peripheral blood mononuclear cells, PBMCs) just above the 1.077 g/mL layer. The PBMC phase was collected, and the leftover erythrocytes were lysed using an ammonium chloride solution by hemolytic shock. After centrifugation (1500 rpm, 5 min at 25 °C), the pellet was washed with RPMI, centrifuged twice, and then suspended in 5 mL of supplemented RPMI (FBS 10%, gentamicin 50 µg/mL, and Gln 2 mM). The PBMC preparation was then adjusted to 1.10^6^ cells/mL for assays.

### 4.5. Human Monocytic Leukemia Cells

The human monocytic leukemia cell line THP-1 (American Type Culture Collection ATCC, TIB-202™) was cultured and propagated at 37 °C in a humidified atmosphere of 5% CO_2_ in RPMI supplemented with 10% FBS, 2 mM Gln, and 50 µg/mL gentamicin. THP-1 cells (4 × 10^5^ cells/mL) were incubated in 6-well plates in a complete growth medium (RPMI) containing 16.2 nM phorbol 12-myristate 13-acetate (PMA, Sigma-Aldrich) for three days for activation into macrophages. Then, they were polarized into M1-like macrophages through incubation with 10 pg/mL of LPS (Sigma-Aldrich) and 20 ng/mL of IFNγ (Gibco) for 24 h.

### 4.6. Kinetics of ROS Production by Leukocytes

Leukocyte preparations (n = 3) were obtained as previously described. The cells (10^6^ cells/mL) were placed in 96-well plates and incubated with APG (0, 10, 25, 50, 100 µg/mL) and dihydrorhodamine 123 (Dhr 123, 1 µM, Sigma-Aldrich), and stimulated (or not) by 1 µM PMA for 120 min. The fluorescence intensity of rhodamine 123, which is the product of Dhr 123 oxidation by ROS, was recorded every 5 min for 120 min (excitation/emission: 485/535 nm) using the Tecan Spark^®^ (Männedorf, Switzerland) [57]. Results were expressed as the percentage of ROS production of the stimulated treated cells compared to the stimulated untreated cells (100%).

### 4.7. Leukocyte Viability

A suspension of 10^6^ cells/mL (in RPMI supplemented with FBS 10%, gentamicin 50 μg/mL, and Gln 2 mM) was placed in 96-well plates incubated with APG at four different concentrations (10, 25, 50 or 100 µg/mL), PMA (0 or 1 µM), and resazurin (25 µg/mL) [57]. Fluorescence (excitation/emission: 544/590 nm) was recorded after 2 h using the Tecan Spark^®^. The results are expressed as the percentage of cell viability of the stimulated treated cells compared to the stimulated untreated cells (100%).

### 4.8. DPPH Assay

One hundred microliters of workable solutions (dilutions of dry extract in ethanol) were mixed with 3.9 mL of DPPH reagent (0.18 g/L in ethanol) and maintained in complete darkness for 30 min at room temperature. The absorbance at 515 nm was measured, and the effective concentration value (EC50) was calculated, which is the concentration of a sample that results in a 50% reduction in the initial DPPH concentration.

### 4.9. Elisa PGE2

Leukocyte preparations (10^6^ cells/mL) (3 volunteers) were incubated with or without LPS (1 µg/mL) and APG (50 µg/mL) for 24 h. PGE2 in the culture media was assessed by ELISA, using the PGE2 assay kit from R&D Systems (Minneapolis, MN, USA).

### 4.10. Determination of Cytokine Concentrations

PBMCs (10^6^ cells/mL) (3 volunteers) were incubated with or without phytohemagglutinin (PHA, 5 µg/mL) and APG (50 µg/mL) for 24 h and the supernatants were collected to measure cytokine secretions using the Human Custom ProcartaPlex assays (Invitrogen™; Thermo Fisher Scientific). All samples were run in duplicate and were assayed for 10 human cytokines (IFNγ, IL-12 p70, IL-1β, IL-2, IL-23, IL-17a, IL-6, IL-8, MIP-1α, and TNF-α). Cytokine levels were measured using optimal concentrations of standards and antibodies according to the manufacturer’s instructions. After performing all assay steps, plates were read in the Luminex Bio-Plex 200 system (Biorad, Marnes-la-Coquette, France) and data were analyzed using BioPlex Manager™ 4.1 software with five-parameter logistic regression (5PL) curve fitting. The results were presented as the percentage between the stimulated treated cells compared to the stimulated untreated cells (100%).

### 4.11. Real-Time Quantitative PCR (RT-qPCR)

Blood leukocytes were incubated with or without LPS (10 µg/mL) and APG (0 or 50 µg/mL) for 4 h. Total RNA was extracted with TRIZOL reagent (Invitrogen, Thermo Fisher Scientific). After the evaluation of the quantity and purity (Tecan Spark^®^), Dnase treatment (Dnase I Amplification grade, Invitrogen), and cDNAs retrotranscription (HighCap cDNA RT Kit RNAse inhib, Invitrogen) were performed according to the manufacturer’s recommendations. Amplification reaction assays were performed using SYBRGreen PCR Master Mix (Life Technologies, Thermo Fisher Scientific) and primers (Table 1) on StepOne^TM^ (Life Technologies). The expression of the following genes was measured: *GAPDH*, *IL-1β*, *TNF-α*, *IL-6*, *IL-8*, *CXCL10*, *NLRP3*, *CASPASE1*, *Nrf2*, *GPX1*, *HO-1*, and *COX2*. Genes were considered significantly expressed and their transcript measurable if their corresponding Ct value was less than 35. Each sample was normalized to the endogenous reference gene (*GAPDH*). The relative quantification method (RQ = 2^−ΔΔCT^) was used to calculate the relative gene expression of given samples with ΔΔCT = [ΔCT (sample 1) − ΔCT (sample 2)] and ΔCT = [CT(target gene) − CT(reference gene)].

### 4.12. Quantitative Measurement of NF-κB by ELISA

PBMCs (10^6^ cells/mL) (4 volunteers) were incubated with or without LPS (10 µg/mL) and APG (50 µg/mL) for 2 h. Pellets from treated cells were collected for the measurement of NFκB phosphorylation using the NFκB p65 (Total/Phospho) Human InstantOne™ ELISA kit (Thermo Fisher Scientific). Absorbance was measured at 450 nm, and the phospho-NFκB p65 over total-NFκB p65 ratio was then calculated. Data are expressed in percentage relative to the control and compared to the positive control group (cells treated with LPS without extract).

### 4.13. Flow Cytometry

Macrophages were detached from the surface of the cell plate using 8 mM PBS/EDTA and incubated under agitation for 35 min at 4 °C. The remaining cells were detached through flushing. Cells were first stained for viability using Viobility Dye 405/520(130-130-404) for 10 min at 4 °C. Then, cells were stained with 4 μL CD14-PE-VIO-770-Human (130-110-521) and 4 μL CD80-PE-Human-REA661(130-123-253) in a total volume of ∼100 μL for 15 min at 4 °C and 15 min at room temperature. After staining, cells were washed once with PBS (without Ca^2+^ and Mg^2+^), centrifuged at 300× *g* for 5 min at RT, resuspended in 350 µL of PBS, and then placed at 4 °C for acquisition. Samples were acquired within 1 h of storage. Flow cytometry was performed after the optimization of the panel using a BD-LSRII flow cytometer (minimum 20,000 live cells counted) in order to check macrophage activation (M0 %) and polarization (M1 %). Instrument setup and performance tracking were performed daily using instrument-specific Cytometer Setup and Tracking (CS&T) beads (BD) using the CS&T program. Results were analyzed with FACSDiva version 9.1 software (BD Biosciences, Franklin Lakes, NJ, USA). Results are presented as a histogram using the total number of labeled cells as the total number of cells.

### 4.14. Statistical Analysis

All the experiments were performed 3–5 times. Values are shown as mean ± SEM. Data were analyzed using one-way ANOVA analysis followed by Dunnett’s *t*-test in GraphPad Prism software version 8 (GraphPad Software, San Diego, CA, USA). Statistical significance between two groups was evaluated using Student’s *t*-test. *p* < 0.05 was considered significant.

## Figures and Tables

**Figure 1 ijms-25-00416-f001:**
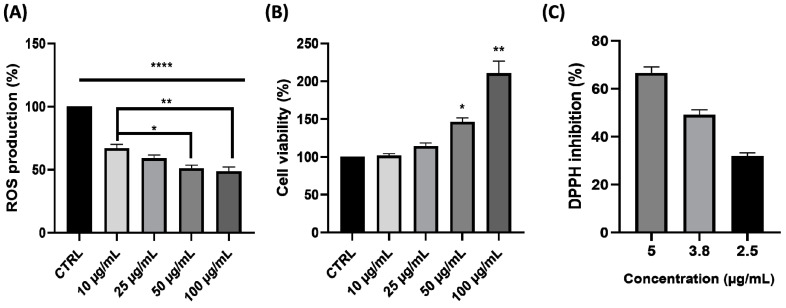
Effect of *Ampelopsis grossedentata* extract (APG) on ROS production *via* blood leukocytes and their viability. Cells were incubated with APG (0, 10, 25, 50, 100 µg/mL) and stimulated with PMA (1 µM). After 2 h, we measured ROS production (**A**), cell viability by the resazurin test (**B**), and free-radical scavenging activity toward DPPH radicals (**C**). Data are expressed as mean ± SEM (Control = 100%) and analyzed using one-way ANOVA, followed by Dunnett’s post hoc test (n = 3) (versus CTRL). Differences were considered significant at *p* < 0.05. * *p* < 0.05, ** *p* < 0.01, **** *p* < 0.0001.

**Figure 2 ijms-25-00416-f002:**
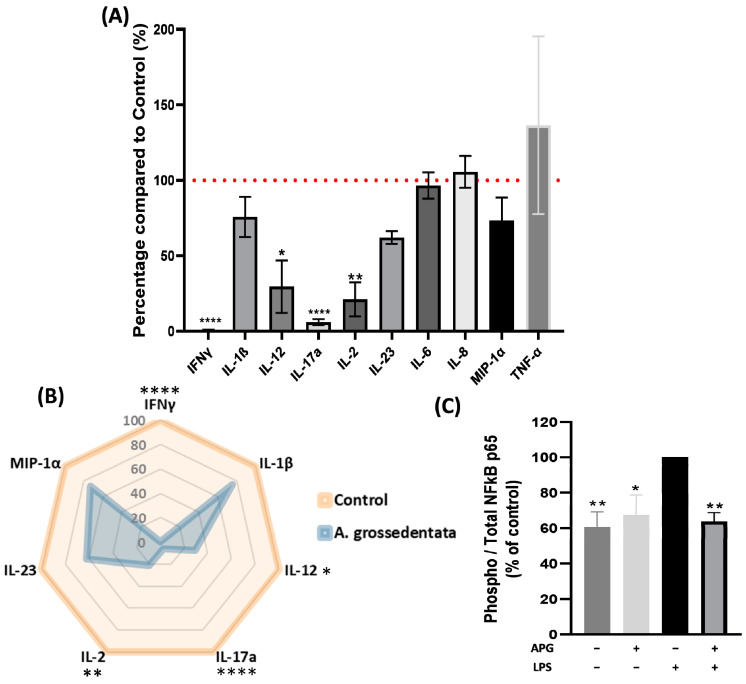
Effect of *Ampelopsis grossedentata* extract on stimulated PBMCs. (**A**) Cells were incubated with or without PHA (5 µg/mL) and APG (0 or 50 µg/mL) for 24 h. Cytokine levels were measured with the Luminex Bio-Plex 200 System using optimal concentrations of standards and antibodies according to the manufacturer’s instructions. Red dotted line corresponds to the 100% normalized positive control. (**B**) Visual representation of pro-inflammatory cytokine production. (**C**) Cells were incubated with or without LPS (10 µg/mL) and APG (0 or 50 µg/mL) for 2 h. Pellets from treated cells were collected, and Nf-κB phosphorylation was measured using the NFκB p65 (Total/Phospho) Human InstantOne™ ELISA kit. Data are expressed as mean ± SEM (Control = 100%) and analyzed using Student’s *t*-test (n = 3–4) (versus CTRL). Differences were considered significant at *p* < 0.05. * *p* < 0.05, ** *p* < 0.01, **** *p* < 0.0001.

**Figure 3 ijms-25-00416-f003:**
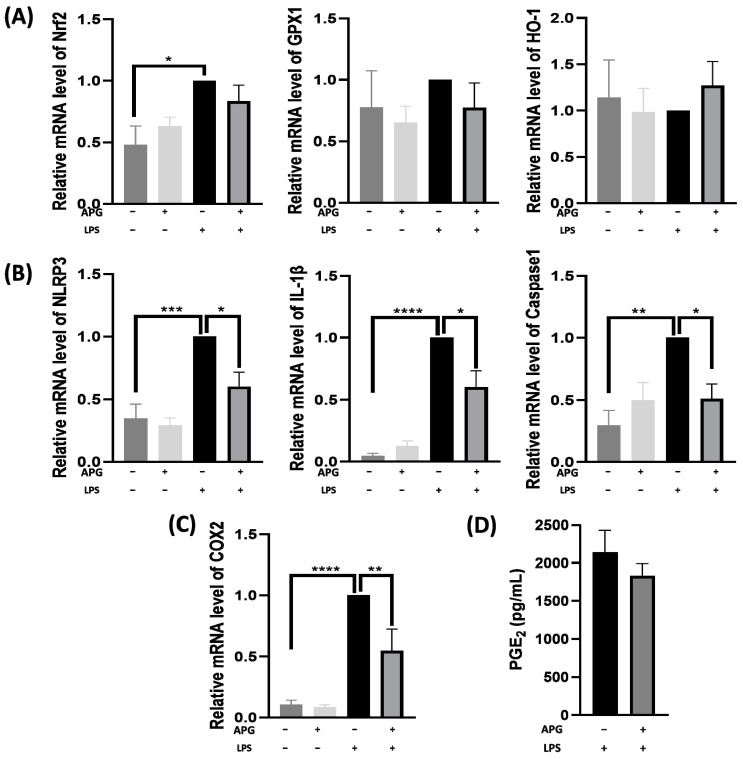
Effect of *Ampelopsis grossedentata* extract on the antioxidant and inflammatory response in blood leukocytes. Cells were incubated with or without LPS (10 µg/mL) and APG (0 or 50 µg/mL) for 4 h. The expression level of these genes was quantified using real-time qPCR and normalized using *GAPDH* as an internal control. (**A**) Antioxidant-sensitive gene expression. (**B**) Expression of NLRP3 inflammasome effector genes. (**C**) Cells were treated as described before, and *COX-2* gene expression was measured using real-time qPCR. (**D**) The secretion of PGE2 in the culture media was assessed by ELISA after 24h incubation. Data are expressed as mean ± SEM and analyzed using one-way ANOVA followed by Dunnett’s post hoc test (n = 4). Differences were considered significant at *p* < 0.05. * *p* < 0.05, ** *p* < 0.01, *** *p* < 0.005, **** *p* < 0.0001.

**Figure 4 ijms-25-00416-f004:**
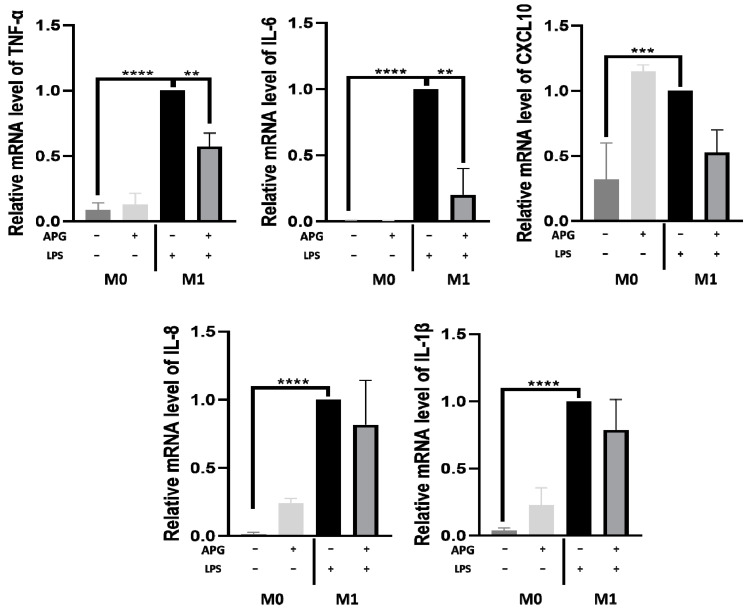
Effect of *Ampelopsis grossedentata* extract on macrophage polarization. THP-1 cells were supplemented with PMA (10 µM) to allow their activation and incubated in a complete growth medium containing LPS and IFNγ for polarization into the M1-type. The treatment with APG (50 µg/mL) was evaluated at each step. Gene expression was quantified using real-time qPCR and normalized using *GAPDH* as an internal control. Data are expressed as mean ± SEM and analyzed using one-way ANOVA followed by Dunnett’s post hoc test (n = 3). Differences were considered significant at *p* < 0.05. ** *p* < 0.01, *** *p* < 0.005, **** *p* < 0.0001.

**Figure 5 ijms-25-00416-f005:**
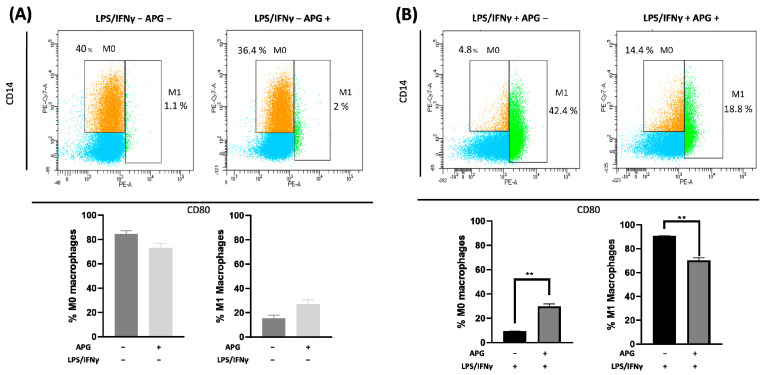
Effect of *Ampelopsis grossedentata* extract on macrophage polarization. (**A**) The polarization of macrophages was analyzed using flow cytometry. The representative side scatter area plots are shown for CD14 (M0) and CD80 (M1) labeling. The percentage of M0 (CD14+) and M1 (CD14+ CD80 + and CD14- CD80+) macrophages in each group using the total number of labelled cells as the total number of cells. M0-like condition. (**B**). M1-like condition. Data are expressed as mean ± SEM and were analyzed using one-way ANOVA followed by Dunnett’s post hoc test (n = 3). Differences were considered significant a *p* < 0.05. ** *p* < 0.01.

**Figure 6 ijms-25-00416-f006:**
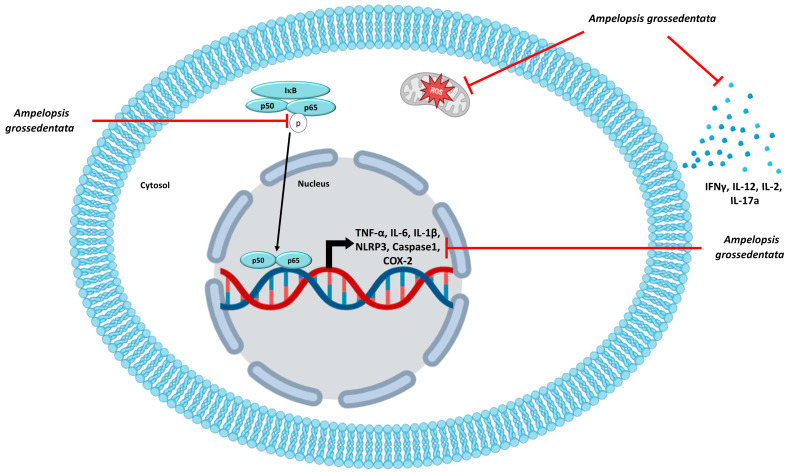
Anti-inflammatory mechanisms of APG extract. IL-2: interleukin-2; IL-6: interleukin-6; IL-12: interleukin-12; IL-17a: interleukin-17a; IL-1β: interleukin-1β, IFNγ: interferon gamma; TNF-α: tumor necrosis factor; COX-2: cyclooxygenase-2; NLRP3: NLR Family Pyrin Domain-Containing 3; ROS: reactive oxygen species.

**Table 1 ijms-25-00416-t001:** PCR primer sequences.

Gene	Species	Forward Primer Sequence (5′-3′)	Reverse Primer Sequence (5′-3′)
*GAPDH*	Human	CACATGGCCTCCAAGGAGTAA	TGAGGGTCTCTCTCTTCCTCTTGT
*IL-8*	Human	CTGGCCGTGGCTCTCTTG	CCTTGGCAAAACTGCACCTT
*IL-1β*	Human	CCTGTCCTGCGTGTTGAAAGA	GGGAACTGGGCAGACTCAAA
*IL-6*	Human	GCTGCAGGCACAGAACCA	ACTCCTTAAAGCTGCGCAGAA
*TNF-α*	Human	TCTTCTCGAACCCCGAGTGA	GGAGCTGCCCCTCAGCTT
*CXCL10*	Human	GGAAATCGTGCGTGACATTA	AGGAAGGAAGGCTGGAAGAG
*COX-2*	Human	CCCAGGGCTCAAACATGATG	TCGCTTATGATCTGTCTTGAAAAACT
*NLRP3*	Human	CCACAAGATCGTGAGAAAACCC	CGGTCCTATGTGCTCGTCA
*Caspase1*	Human	GCCTGTTCCTGTGATGTGGAG	TGCCCACAGACATTCATACAGTTC
*HO-1*	Human	ACAGTTGCTGTAGGGCTTTA	CTCTGAAGTTTAGGCCATTG
*GPX1*	Human	GCACCCTCTCTTCGCCTTC	TCAGGCTCGATGTCAATGGTC
*Nrf2*	Human	CACATCCAGTCAGAAACCAGTGG	GGAATGTCTGCGCCAAAAGCTG

## Data Availability

The data presented in this study are available on request from the corresponding author.

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
