# Peer review of "Exploring the Therapeutic Potential of Ampelopsis grossedentata Leaf Extract as an Anti-Inflammatory and Antioxidant Agent in Human Immune Cells"

_ijms, 2023, doi:10.3390/ijms25010416_

Round 1

Reviewer 1 Report

Comments and Suggestions for Authors

The authors presented a very interesting and detailed study of Ampelopsis grossedentata, however, the authors should take below notes into consideration. It is important that all the figures are legible, that all of the latest research on the topic is properly cited and discussed:
1) the font size is too small in the figures; the authors should maintain the same font type across all the figures; lines 210-212: unusual highlight.
2) The authors should avoid nonspecific statements like "reduced levels", "greatly lowered", and instead specify how much.
3) The authors should include a review of the latest papers on the Ampelopsis grossedentata and discuss how this paper is differentiated from them:
Wang et al., Antioxidants and Li et. al. RSC Advances

4) The authors should further elaborate on the limitations of this study as well as future directions.

Author Response

Reviewer 1

All the modifications were highlighted in yellow in the manuscript

  1. The font size is too small in the figures; the authors should maintain the same font type across all the figures; lines 210-212: unusual highlight.

Response to reviewer 1:

According to the reviewer, we have checked all the figures. We modified the font size of the legends (Palatino 10, same size as manuscript text). We have also checked the size of the axis. Figure 4 has now been split into 2 figures (Figures 4 and 5) to maintain a readable font size.

The unusual highlight has been suppressed lines 210-212 corresponding now to the lines 240-243.   

  1. The authors should avoid nonspecific statements like "reduced levels", "greatly lowered", and instead specify how much.

Response to reviewer 1:

We're a little disappointed by this comment. Indeed, we have always specified variations in modifications with precise data in the section Results.

Examples:

-Lines 115-117: “Treatment with APG induced a significant decrease in IFNγ, IL-12, IL-17a, and IL-2 (-99.3±0.6%, -61.5±20.4%, -94±3.6%, and -79.8±14% respectively) compared to the control (Figure 2A). For IL1-β, MIP-1α and IL-23, a non-significant decrease was observed (-24.3±16.5%, -27.2±20.5% and -38±5.2%, respectively). The treatment did not seem to have any impact on IL-6, IL-8 and TNF-α secretion by PBMCs.”

- Lines 126-127: “The treatment with APG on LPS-stimulated PBMCs induced a significant decrease in NF-κB p65 phosphorylation (-36.3±10.4%).”

- Lines 172-175:  “As expected, LPS significantly induced the expression of NLRP3, IL-1β and Caspase-1 (Figure 3B), while APG greatly decreased the expression of all the studied genes (RQ=0.60±0.23; 0.60±0.26 and 0.51±0.20 for NRLP3, IL-1β and Caspase-1, respectively).”

But according to the reviewer’s comment, we've included some more data in the discussion section.

-Lines 266-267: “In our investigation, we observed that APG lowered NF-κB pathway activation by decreasing the phosphorylation of the p65 subunit by 36 % (Figure 6).”

-Lines 276--278: “Our study revealed in PHA-stimulated PBMCs that APG inhibited the production of several inflammatory cytokines: IFNγ, IL-12, IL-17a, IL-2 by 99%, 61 %, 94%, and 80% respectively (Figure 6)”

-Lines 326-327: “Our findings revealed that APG greatly lowered by 20 % the polarization of M1 macrophages.”

  1. The authors should include a review of the latest papers on the Ampelopsis grossedentata and discuss how this paper is differentiated from them: Wang et al., Antioxidants and Li et. al. RSC Advances

Response to reviewer 1:

According to the reviewer’s recommendations, we added the papers of Wang et al. in the section discussion concerning the results. Indeed, they obtained interesting results in their in vitro studies and in vivo experiments. So we added these data in the discussion:

-Lines 285-287: “In another study, which is closer to ours in terms of evaluated extract, Wang et al showed a decrease in the quantity of IL-6 and IL-1β (50 mg/kg for IL-6 and 100 mg/kg for IL-1β) LPS-stimulated ICR mice serum”.

-Lines 342-347: “Two other studies carried out on extracts of Ampelopsis grossedentata showed divergent results probably due to the extraction method. Indeed, the first study conducted by Zhang and al, on vine tea powders extracted with petroleum ether showed an IC50 of 15.35 µg/mL, which is similar to our results [52]. On the other hand, Wang et al, showed an IC50 of around 0.4 mg/mL for a commercial water extract of Ampelopsis grossendentata [33].”

On the contrary, we decided not to include the paper of Li et al. as they focused on anticancer and antibacterial activities and not at all on anti-inflammatory activities.

During the revision process, we found 2 new reviews on Ampelopsis grossedentata in October 2023 (10.3390/molecules28207145) and december 2023 (10.1016/j.jep.2023.116788). if the reviewer considers it necessary to add them, we can include them in the manuscript.

  1. The authors should further elaborate on the limitations of this study as well as future directions.

Response to reviewer 1:

At the end of the discussion, we set out the limitations of our study. We believe that the greatest limitation of our work is that we are directly testing the impact of the extract on cultured cells. However, given the low bioavailability observed in other studies, the effective in vivo concentrations would be lower.

-Lines 364-376: Nonetheless, the study is limited by the question of bioavailability and adequate dos-age. Indeed there have been some bioavailability and pharmacokinetic research on DHM, but none on Ampelopsis grossendatata extract and DHM investigations demon-strated very poor bioavailability. Indeed, Ruan et al, demonstrated in an in vitro investigation utilizing Caco-2 cells that the molecule's bioavailability was less than 10% [56]. Furthermore, Liu et al, showed in in vivo investigation in rats that the maximal concentration (Cmax) of DHM was only 21.63 ± 3.62 ng/mL after 2.67 h of treatment with an oral dosage of 20 mg/kg. After intravenous administration of 2 mg/kg DHM, the Cmax was 165.67 ± 16.35 ng/ml, the half-life was 2.05 ± 0.52 h, and the residence time was only 2.62 ± 0.36 h, according to the same study. In rats, bioavailability was found to be 4.02% [27]. To replicate the in vitro effects, physicochemical characteristics of this molecule would need to be considerably modified in order to make it more bioavailable and hence enhance its effects in vivo.

Given the anti-inflammatory activities found in our study, Ampelopsis grossedentata could appear to be a promising option for managing inflammation. So it would seem interesting to identify the other major constituents of the extract in addition to DHM. Furthermore, it will be essential to carry out investigations seeking to improve bioavailability.

Thus, we decided to add the following sentence (Lines 379-380): “Further studies are required to determine the other main components of our extract and to investigate bioavailability assays”.

Reviewer 2 Report

Comments and Suggestions for Authors

Chervet A. et. al. in this paper have explored the role of Ampelopsis grossedentata leaf extract as an anti-inflammatory phytochemical. The authors treated PBMCs and macrophages with varying doses of APG extract and saw decrease in various pro-inflammatory cytokines. The reviewer feels the authors need to make following major revisions to make it a better read.

              Major Points:

·      Fig1 Was similar decrease in ROS levels were also seen in macrophages? Reviewer suggests doing these experiments in macrophages as well.

·      Fig2 and Fig3 can be combined.

·      Fig4 Why was there no difference Nrf2 and its downstream gene levels given they have been shown to increase in literature. Probably a longer treatment is required to see quantifiable changes.

·      Reviewer suggests the authors perform a WB for check change in protein expression of these genes.

·      Figure 6 seems misleading, APG might have been shown to decrease inflammation by these mechanism in the literature, but it was not shown in this particular paper.

Minor Points:

·      There are two 2.1 sections.

              Line 93DMH? Is it DHM?

Author Response

Reviewer 2

All the modifications were highlighted in yellow in the manuscript.

Major Points:

  1. Fig1: was similar decrease in ROS levels were also seen in macrophages? Reviewer suggests doing these experiments in macrophages as well.

Response to reviewer 2:

While the formation of reactive oxygen species (ROS) by macrophages—especially NO°—is intriguing we opted to use total leukocytes and follow the generation of ROS from total leukocyte circulating cells, specifically monocytes and PNNs, which are the principal ROS-producing cells.

However, as macrophages are the cells that are found in tissues during an inflammatory state, we decided to investigate the effect of our extract on the polarization of monocytes into pro-inflammatory macrophages  in this study.

  1. Fig2 and Fig3 can be combined.

Response to reviewer 2:

As suggested by the reviewer, we combined Fig 2 and Fig3 into one figure (Figure 2).

  1. Fig4 Why was there no difference Nrf2 and its downstream gene levels given they have been shown to increase in literature. Probably a longer treatment is required to see quantifiable changes.

Response to reviewer 2:

Our results seem to be consistent since we didn't find significant modifications in the expression of neither NRF2 nor the downstream genes.

Indeed, some authors found gene level modifications of this biological pathway. However, the cell lines employed in the two investigations cited in our manuscript (doi:10.1007/s10495-017-1381-3 and doi:10.1002/biof.1395) were endothelial cells (HUVECs). The stimulating agents were also different (palmitic acid or ox-LDL). Finally, while they used DHM, we tested extract from Ampelopsis grossedentata. The differences observed between our studies could therefore be attributable to these different characteristics.

We agree with the reviewer. A longer treatment may have a stronger effect.

NB: There was significant variance between CTRL-negative and CTRL-positive cells, with LPS enhancing NRF2 expression in the former. The star has been omitted and has been now added (Fig 3A).

  1. Reviewer suggests the authors perform a WB for check change in protein expression of these genes.

Response to reviewer 2:

We were unable to conduct a western blot analysis on our entire leukocyte cell population. Actually, these WB analyses necessities an extensive number of cells, and we only have a limited cell number

As Nf-κB is a key actor of inflammation, we evaluated it as a priority by an Elisa assay which allows to  use less cells.

 Figure 6 seems misleading, APG might have been shown to decrease inflammation by these mechanism in the literature, but it was not shown in this particular paper.

Response to reviewer 2:

We thanks the rewiever for this recommendation and the figure has been now modified in the aim to clarify the impact of the Ag extract. Indeed, NLRP3, IL1b, Caspase1 and Cox2 have been incorporated at the transcriptional level and are no longer present in the cytoplasm. Furthermore, the cytokines on which the extract had an influence have been included. Finally, the links between ROS- Nf-κB, ROS-Nrf2, and ROS-Inflammasome (NLRP3) have been removed because they are not discussed in the article.

 Minor Points:

  1. There are two 2.1 sections.

Response to reviewer 1:

Sorry for this misspelling which has been now corrected (line 92). 

  1. Line 93DMH? Is it DHM?

Response to reviewer 1:

This is a misspelling, we modified it page line 93

Round 2

Reviewer 2 Report

Comments and Suggestions for Authors

Accept in present form.